# Voices in Shaping Water Governance: Exploring Discourses in the Central Rift Valley, Ethiopia

Amare Bantider [1,2,*], Bamlaku Tadesse [1], Adey Nigatu Mersha [1], Gete Zeleke [1], Taye Alemayehu [1,3], Mohsen Nagheeby [4] and Jaime Amezaga [4]

1   Water and Land Resource Centre, Addis Ababa University, Addis Ababa P.O. Box 3880, Ethiopia
2   College of Development Studies, Addis Ababa University, Addis Ababa P.O. Box 3880, Ethiopia
3   Ethiopian Water Resources Institute, Addis Ababa University, Addis Ababa P.O. Box 3880, Ethiopia
4   Water Security and Sustainable Development Hub, School of Engineering, Newcastle University, Newcastle NE1 7RU, UK
*   Correspondence: amare.b@wlrc-eth.org or amare.bantider@aau.edu.et

**Abstract:** As is the case elsewhere in the world, water governance in Ethiopia is a by-product of a complex set of various global and local socio-political, economic, and ecological discourses and narratives. However, the many competitive and often conflicting discourses on shaping water governance in the Ethiopian Central Rift Valley (CRV) have not been examined and chronicled. This paper investigates the different discourses, narratives, and debates of water governance and their implications for satisfying the growing demand for water. The study was grounded in political economy and political ecology theoretical frameworks. Data were collected through literature surveys and intensive fieldwork, and were analyzed following a discourse analysis and using narrative analysis techniques. The study found that the dominant competing discourses that have greatly influenced water governance in the CRV focus on decentralization, water-centered development, marketization, land/water degradation, climate change, water scarcity, and weak water governance. We suggest that the analysis and documentation of the diverse narratives and discourses from multiple perspectives could help to unravel the complex nature of water governance in the CRV and lay the foundation for attempts to implement sustainable water resource management in the region.

**Keywords:** water governance; competing discourses and narratives; water demand; Central Rift Valley; Ethiopia

## 1. Introduction

The perpetual need for an adequate supply of safe water to aid a decent living standard is unquestionable. This need is growing rapidly for both domestic use and economic development (agriculture, energy, and industry). For many developing countries, water resource development and management remain at the heart of efforts to achieve sustainable development, economic growth, and poverty reduction [1]. At the same time, there is growing competition between different types of water usage, including environmental flow and water users [2]. However, most developing countries have yet to adequately adapt their water infrastructure and institutions to meet the rising demand. In Ethiopia, the unmet demand and challenges of water resource development and management are diversified by a multitude of factors and drivers, at the core of which is weak governance.

The concept of water governance comprehends complex and contested political, economic, and social processes and institutions by which governments, civil society, and the private sector make decisions about using, developing, and managing water resources [3,4]. According to studies conducted in many countries, weak water governance is a major problem that hinders the optimal implementation of integrated water resource management (IWRM) [5–7].

Discourses, narratives, and debates play impelling roles in policy design, decision-making, planning, and water governance. They can even serve as "instruments of power" when implementing political objectives [8–11]. They are also, in turn, influenced by institutional as well as personal practices. Specifically, Hajer and Versteeg argue that discourses are a product of institutional practices and individual activities that reflect particular types of knowledge. They defined it as "an ensemble of ideas, concepts and categories through which meaning is given to social and physical phenomena, . . . produced through an identifiable set of practices" [11] (p. 175). Robinson and Crane use discourses and narratives almost interchangeably, and they explain discourses or narratives as "institutionalized linguistic and narrative frames that shape actors' interpretations of information, as well as inform their action choices" [9] (p. 4). In agreement with but a bit different from the above, Kleinschroth et al. define narratives as widely understood streams of argumentation and meaning-making to social processes that play a strong role in the policy process, including in the framing of research on development and sustainability transitions and in environmental decision-making [8] (p. 1859). In this paper, they are used interchangeably based on the preceding definitions and understandings of the two terms. Accordingly, collating the varied water management-related discourses provides several insights that could smooth the way for planners and decision-makers to thoroughly examine a pull of perspectives on water management and development as part of the endeavor to achieve sustainable water governance and thereby help to ensure water security for all. Among the widely debated discourses and narratives are: water scarcity and crises, marketization, privatization, and participation [5]; water security and water supply inequality [12,13]; supply- and demand-side management, deep ecology, and governance of water resources [14]; and integrated water resource management (IWRM), water resource investment, water wars, and water sovereignty [15,16].

While there are many more discourses and narratives on water governance in general, in Ethiopia, scholarly works that examine such discourses are generally scarce and fragmented. To redress this deficiency, we study discourses and narratives concerning water resources to identify the dominant ones and examine how they have influenced policies. This calls for a comprehensive collation and consideration of the myriad arguments from various perspectives to comprehend the water management and governance issues as understood by different parties with concerns in the matter. In this paper, we present and discuss the evolution and existing discourses and narratives on water resource development in the Central Rift Valley (CRV) sub-basin. In addition, an attempt is made to link the water management discourses and narratives with the wider natural resource management of this sub-basin.

The selection of CRV as a case study site is due to several reasons. A few of them are: first, it is a closed (endorheic) lake basin in the great East African Rift Valley system; second, it is among a few areas where modern water resource development in Ethiopia began; and third, its proximity to major urban areas of the country, including the capital city Addis Ababa, and a good transport network linking the CRV to foreign and domestic market outlets attract a large number of competing water users. This enables them to play a role in influencing water resource management. In addition, the prevailing climate change affects the livelihoods of the residents. In general, CRV has very complex settings linked to water resource management and development. Therefore, the study findings from this sub-basin could inform situations in other kinds of lake regions.

This paper is organized into five sections, including a brief introductory one on the water governance system and the situation of the CRV, followed by a review of the main theories and a presentation of the methods employed for the study. It then identifies and discusses different discourses, debates, and narratives, and in the last section, it presents the effects of these discourses and narratives on the CRV's water governance systems. The paper closes with concluding remarks.

## 2. Underpinning Theories and Methods for Framing Competing Narratives, Discourses and Debates

### 2.1. The Central Rift Valley (CRV) Context

The CRV is one of the sub-basins of the Rift Valley Basin in Ethiopia, which is part of the Greater East African Rift System (Figure 1). It is situated in central Ethiopia, located between 7°00′–8°30′ N and 38°00′–39°30′ E, and shared by the Oromia Region and the Southern Nations, Nationalities, and Peoples' Region (SNNPR). With a total area of about one million hectares (10,000 km$^2$), the sub-basin covers five administrative zones and 30 woredas (in Ethiopia, woreda, an equivalent to district, is the second-smallest administrative unit) in these two regional states. The CRV encompasses four large lakes: Lake Ziway, Lake Abijata (Abiyata), Lake Langano, and Lake Shalla. It also has rivers such as the Bulbula, Meki, and Katar. The lakes are fed by interconnected streams flowing from the western and eastern escarpments. The Central Rift Valley plays a vital role in the country's social, economic, and ecological systems. Furthermore, the diverse topographical conditions of the basin, with elevations ranging from about 1550 to 4200 m above sea level (masl), give rise to diverse ecosystems and biomes, including the lake ecosystems, lowlands dominated by acacia species, a mid-altitude tropical montane forest belt, and sub-afro-alpine and afro-alpine belts, all of which give rise to the occurrence of several agro-climatic belts. It, therefore, hosts unique flora and fauna biodiversity (see Figure S3). In particular, it is recognized worldwide for its diversity of bird species [17].

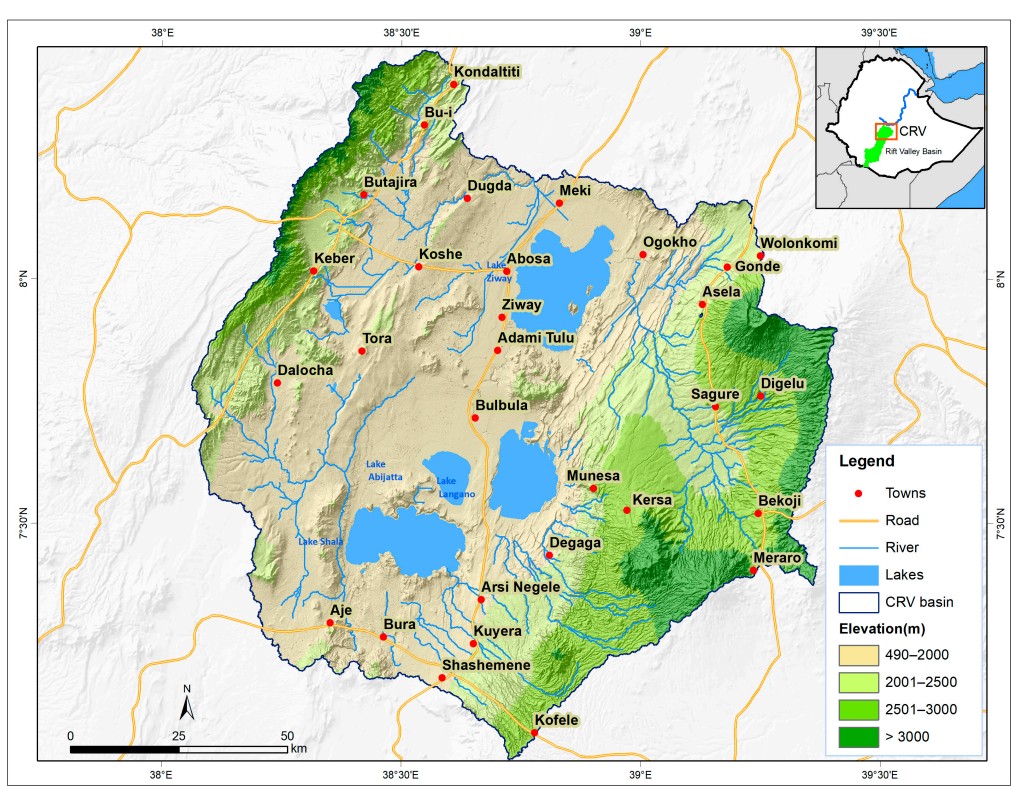

**Figure 1.** Location map of the Central Rift Valley sub-basin, Ethiopia.

The dominant livelihood system in the CRV basin is mixed agriculture (crop and livestock production). Historically, lowlanders were pastoralists (now mixed farmers), whereas the mid- and highlanders practice mixed rain-fed agriculture. In recent years, irrigation-based agricultural production systems, including commercial horticulture (flowers, fruits, and vegetables) production, small-scale vegetable farming, extensive irrigated cereal crop farming, and various industries (agro-industries, soda-ash manufacturing, wineries, etc.) flourished in the area. According to the data gathered from the respective woredas, the total population of the basin in 2021 is estimated at 3.2 million, with the highlands being

more densely populated relative to the lowlands. The area is predominantly inhabited by the Oromo, Siltie, Guraghe, and other ethnic groups in the basin.

The CRV's inherently diverse geography and dynamic socio-cultural-political settings have driven the evolution of diverse water governance arrangements and reorganizations over the past several decades. The related features of the socio-ecological context—as reflected in the policy provisions, legal and regulatory frameworks, norms, institutional arrangements, power distributions, and complex networks and interactions of stakeholders—provide the basis for giving insights into water resource governance systems. The insights derived from the analysis of the elements and processes of the governance system can, in turn, be used to illustrate a range of narratives and discourses, which are then employed to assess the system's performance.

### 2.2. Framing the Discourses and Narratives

Discourses and narratives are types of social practices in which various actors exercise their power to serve their interests. Actors may construct and employ discourses in an attempt to make or influence policy options. Such discourses may represent, construct, and transform the social reality of water. Accounts of reality inherently reflect certain dominant discourses and the power dynamics that maintain those discourses [10]. Having such political dynamics at its center, this research adopts political economy and political ecology theories to frame the narratives, discourses, and debates on water/natural resource governance.

The political economy and political ecology theoretical frameworks were used to comprehend various environmental and economic issues as viewed by different political systems and ideologies. According to Swyngedouw, political economic and political ecological perspectives on water suggest a close correlation between the management of water, including transformations of water in its hydrological cycle at multiple spatial and temporal scales and levels, and the intricate relationships of social, political, economic, and cultural powers [18]. Scholars categorize non-human nature as a space of political significance that emerges from competitions among various social actors with political power asymmetries to secure access to and control over natural resources [19–21]. The existence of power asymmetries between different actors influences the distribution of, access to, and control over vital natural resources such as water and land. Political ecology arguments revolve around actors' identification, their power relations, and the institutions governing access to and control over resources—in the case of this study, water resources. In other words, the study explores who has control over the resource, who the key actors are, who has more power and influence, and who are losers in terms of benefit-sharing. Discourses and narratives also vary due to the disciplinary backgrounds and experiences of discourse/narrative holders.

To frame these narratives, we did the following: (a) reviewed relevant literature accessed by an internet search using keywords including water resource management, water governance, water development, land degradation, water scarcity, and competition, in CRV; (b) reviewed policy documents on water, irrigation, agriculture, water users and cooperatives, and related sectors that were enacted and implemented by the successive governments of Ethiopia, including the present one (see Appendix A for policy and laws reviewed); (c) conducted intensive fieldwork and held 15 focus group discussions (FGDs) and 28 key informant interviews (KIIs) with water users (community elders, youth, and women), various experts, and officials; and (d) made extensive observations at the basin on three separate occasions in 2021 and 2022 (see the route map of observations (Figure S1) in the Supplementary Materials linked to this paper). The FGDs and interviews were focused on the nature of water uses, irrigation practices, types of local water users, actors' identifications and their power relationships, and resource scarcity and competition (see Supplementary Materials on major results of FGDs and KIIs in Table S1). Substantial secondary data on different aspects of water use and governance were also collected from thirty woredas in the CRV sub-region.

## 3. Results

### 3.1. Typologies of Discourses and Narratives That Prevailed in the CRV since the 1960s

There are dominant discourses and narratives of water resource management identified as having prevailed in Ethiopia in general and the CRV in particular since the late 1950s (see Figure 2 and Table 1). The late 1950s are considered the beginning of the water discourse, as that was when major hydrological resource potential studies commenced in the country during the periods of the First Five-year National Development Plan (1957–1961) and its successor, the Second Five-year National Development Plan (1962–1967). The types and arrangements of the water governance discourse and narratives are presented here along with the ideologies and the political economic policies adopted by successive Ethiopian governments. Those are the Free-market Economic Policy of the Imperial Government (before 1974), the Command (socialist) Economic Policy of the *Derg (Derg*—is the provisional Military Council that deposed the Emperor Hailieslassies's regime in 1974 through the popular revolution and changed the imperial government's ideology to socialist ideology which reign from 1974–1991), and the Free-market-oriented but mixed economic policies since 1991. In each of these three periods, successive governments adopted specific national plans. The discourses and narratives on natural resource management in general and water resource management in particular were closely related to those ideologies and the accompanied national policies and plans.

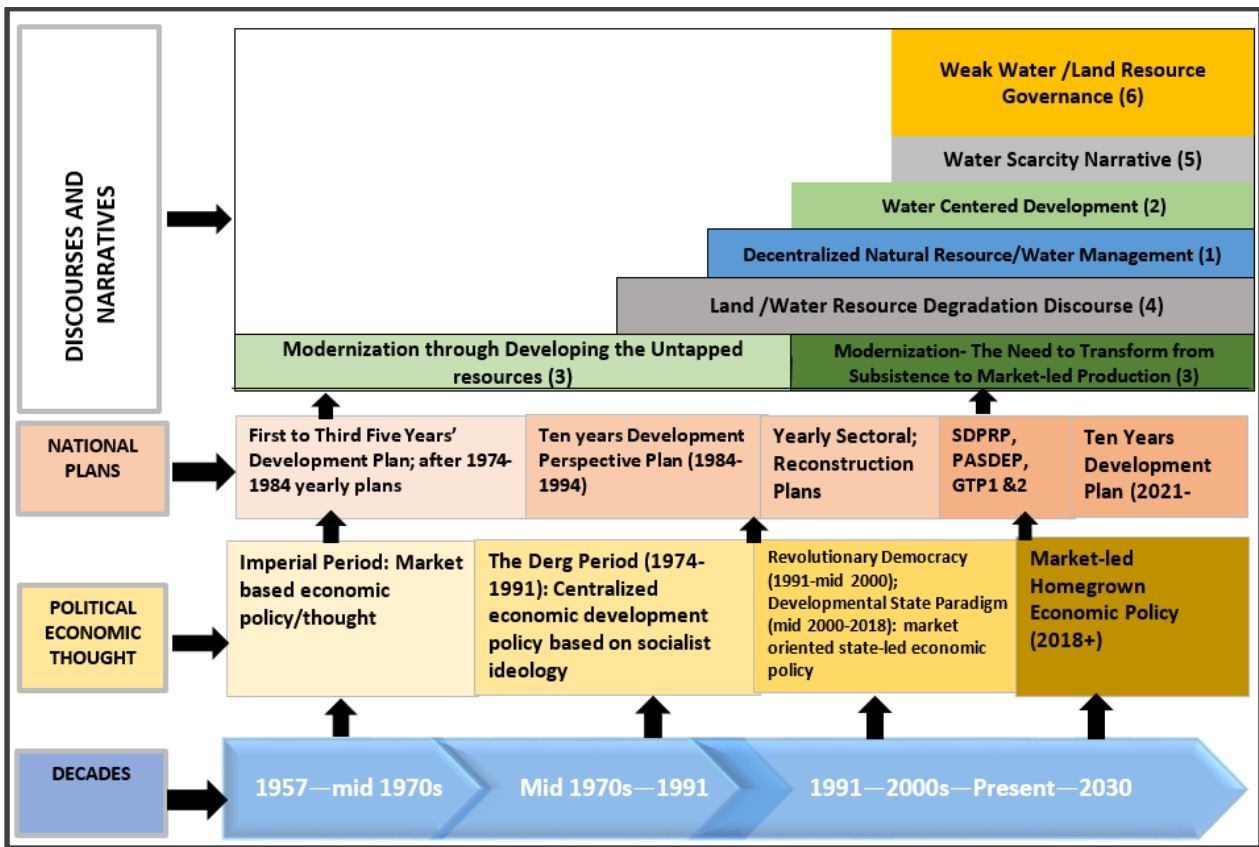

**Figure 2.** A sketch showing a timeline of water resource management discourses and narratives in Ethiopia, which are shaped by different political and economic systems of the period. Note: NR—Natural Resource; SDPRP—The Sustainable Development and Poverty Reduction Program (2002/03–2004/05); PASADEP—Plan for Accelerated and Sustained Development to End Poverty (2005/06–2009/10); GTP 1—Growth and Transformation Plan 1 (2010/11–2015/16); and GTP 2—Growth and Transformation Plan 2 (2015/16–2019/20). The color is to show the different discourses and their emphasis over time. The numbers in the bracket indicate the major discourses discussed in the preceding sub-sections.

**Table 1.** The identified discourses, narratives, and debates on water governance in Ethiopia (1990s to the present).

| Discourse/ Narrative | Owner or Subscriber of the Narrative/ Discourse | Administrative Level of Concern of the Narrative/Discourse (Local to Global) | Implication for Water Governance |
|---|---|---|---|
| The decentralized water resource development narrative (1) | The ruling party, Ministry of Water and Energy (MoWE), and different regional states and their subordinate structures as well as respective political parties. | National, regional, zonal, woreda and *kebele* (*Kebele* is the lowest administrative unit) levels. | Encourages community participation in water governance and promotes equity in distribution and management of water resources. |
| Water-centered development discourse (2) | MoWE, Ministry of Agriculture (MoA); Irrigation Agencies; Cooperatives, Water Users Associations (WUAs), and farmers who use irrigation. | The government at national and regional levels, as well as the diverse water users. | This discourse is brought about by the alarming national food insecurity issues, and, hence, the awareness that the rising demand for food cannot be met by rain-fed agriculture alone. |
| Modernization/market-led development discourse: Transition from subsistence to market-based water resource development (3) | MoWE, MoA, and offices at zonal and woreda levels; NGOs, and local-level water users (individuals and cooperatives). | From local, regional, national, and global levels to ensure food security and sustainable development in the country. | This transition requires huge quantities of water and has implications for water security. |
| Land/water resources degradation and climate change discourse (4) | MoA, MoWE, Environment, Forest, and Climate Change (EFCC), researchers, academicians, and local-level resource users (water). | From global to local level. | It has direct impact on water security since the aggravated environmental degradation negatively affects the present and future water security. |
| The water scarcity narrative (The Aral Sea syndrome in CRV) (5) | Mainly principal water users including local residents, irrigation, or water users' associations, industries/companies (flower farms, agro-industries, and other factories), livestock owners/farmers, and researchers. | The water scarcity problems are the concern of the local, regional, and national experts on water resource management. | The implications of water scarcity narratives to water security are direct, since when water scarcity intensifies, the water security problem at various levels will be aggravated. |
| Week water resource management institutions | Policymakers, practitioners, researchers/academicians, and local-level water users. | This narrative is the concern of all, including policymakers. | It argues that water insecurity and water crisis are the direct outcome of weak institutions and governance problems. |

The political ideology of the Imperial regime was semi-feudal and semi-capitalistic in nature. The dominant economic development policies at the time were free market economic policies where the private sector had a dominant space in economic life. In the latter half of this period, the government developed three successive five-year development plans (FYDPs) to guide the economic development path, including water resource management, starting in 1957 (see Appendix A). As a result, the institutions, including policies and laws, were favoring the construction of hydropower dams, and irrigation-based private commercial agriculture came into the picture. The major objective of irrigation agriculture during that period was to produce highly needed sugar from sugarcane plantations and horticultural crops for the growing urban population and other products, such as cotton and tobacco, to supply raw materials for agro-industries. For example, the Dutch HVA Sugarcane Plantation Estate was established in the 1950s for sugar processing in the Wonji-Shoa and Methara areas; the upper Awash Agro-industry was established in the early 1970s for cotton, tobacco, and horticultural crop production; the Cotton Plantation was set up in the Middle Awash in 1960; Meki-Ziway Irrigation was established in CRV in 1967 (first as a dairy farm and then as horticultural farms); and Melak Sedi-Amibara Irrigation for cotton and banana plantations in 1971 [22]. The major discourse of the period, therefore, revolved around how to use natural resources to modernize the country, fulfill the demands of the urban population, and supply raw materials for the infant agro-industries. Hence, this discourse has ensued to attract investors in the field and thereby provide preference for such modern agriculture, and there were a few questions of competition on water resource use.

During the period from mid-1974 to 1991, the *Derg* followed a socialist ideology. Consequently, the country adopted central planning principles, and hence the role of the private sector in the economy heavily declined. The private irrigation schemes were nationalized by the government, which also established new schemes. The government made efforts to explore the existing water potential in the country through various studies and to use the resources for national growth, i.e., to be self-sufficient in food, produce raw materials for agro-industries, and export for foreign earnings. The dominant discourse in that period was still modernization by investing in natural resource development, but only through public investment.

In this decade, drought and its impact forced the policymakers to invest in irrigation schemes in areas such as the Central Rift Valley basin, where surface water is readily available for development. In addition, the government, for the first time, identified land degradation as a major development challenge (a threat to survival) and embarked on campaigns for the natural resource conservation movement to curb land degradation with a motto of wisely using natural resources. Domestic water supply to urban and rural residents was also on the agenda of the government to satisfy the basic needs of the people based on the socialist principle of equitable distribution of benefits. In addition, fishery development in the CRV's fresh lakes is recognized as an important economic development component. Hence, modernization on the one hand and curbing land degradation on the other were the two major discourses around natural resources/water resources. Equitable economic development was also the goal of the regime.

Since 1991, major political economy ideas shifted from centralized economic development to liberal market-oriented. However, they were substantially led by state and parastatal institutions with a philosophy of "developmental state" principles. In this era, radical and dynamic changes have taken place. The main ones are: (a) a shift in government administrative structure from a "unitary" to a "federal" system; and (b) a shift from centralized economic policy to market-oriented policy. To affect these two major shifts, the government adopted several policy reforms in economic policy to attract the private sector. As a result, large- and small-scale commercial irrigation-based farms, agro-industries, lakeside resorts, and lodges were tremendously expanded in the CRV areas. This policy shift, coupled with the proximity of the CRV to major urban centers of the country, including Addis Ababa, and their accessibility to transport infrastructure (road and rail linking Addis

Ababa and the Djibouti Port and air transport), attracted a large number of actors. The population size and urbanization also expanded in this period.

All these political, administrative, economic, and environmental changes attracted several actors into the basin, with several interests and competitions, leading to the abstraction of large volumes of water per year and other challenges. In parallel, these new developments led to the mushrooming of several discourses and narratives on water resource use in the basin. The dominant discourses include decentralization, water-centered development, market-led water resource development, water scarcity, land/water degradation, competition over water resource use, and weak water governance (please refer to Table 1).

*3.2. Actors, Competing Interests, and Power Relations: Important Factors in Shaping Contemporary Discourses on Water Resource Management in the CRV*

Experts and policymakers commonly say that in the CRV, there are multiple actors with unequal competing interests (Personal communication with the head of the Rift Valley Basin Office, 2021). In the context of the basin, several interdependent factors lead to the genesis of diverse discourses and narratives on water governance. Major factors, based on the data gathered from different relevant sources, include the following:

- New water users due to the flourishing of intensive irrigation both at the individual and cooperative levels, mostly for vegetable production for market; the beginning of the huge influx of tourist and hotel enterprises; the flourishing of flower farm companies (which as of 2021 covered 529 hectares with a water use rate of 7.3 million cubic meters (MCM)); the Castel Winery (453 hectares covered and 1.8 MCM water use rate); the establishment of nine agro-industries (11.8 MCM water use rate); packaged water and different industries; domestic users; and environmental flow requirements;
- The change of land use types and patterns, mainly from rain-fed to irrigation-based mechanized agriculture, such as the introduction of wheat production by irrigation during the dry seasons. According to the Munissa woreda Agriculture Office, about 218 hectares of land were taken up for irrigated wheat production in the 2020/21 production year during the dry season. This was due to the new direction given by the regional government to shift from fruits and vegetables to cereal crops, particularly wheat production, using irrigation in view of achieving local food security and self-sufficiency;
- The rise in water demand results from the increase in human and livestock populations. The human population of the CRV grew from 1.9 million in 1987 to 2.8 million in 2007 and was expected to reach 4.2 million in 2021 [23], an annual growth rate of around 3.6%. The current livestock population in the CRV basin is estimated at more than 9.5 million;
- Changing lifestyles: Many years ago, the lowland parts of the CRV basin were inhabited by pastoralist and agro-pastoralist communities that were heavily dependent on mobile forms of livestock production that were believed to be environmentally friendly. Following the move from a pastoral production system to sedentary agriculturalists, the communities' lifestyle has also changed fundamentally in respect of their dietary system (i.e., a major shift from the consumption of animals and animal products to cereals and cereal products), which is hugely dependent on plow farming. This lifestyle change enlarges the use of water resources in the basin to a degree believed to negatively affect the existing production systems and the natural ecosystems [24];
- There are conflicts between the upper stream and downstream water users at different rivers, among them the Akamuja, Katar, Bulbula, Yagullo, Hulluka, and Tullu-Dema rivers. One cause of such conflicts is the diversion of rivers and the abstraction of water by pumps upstream that minimize the volume of water flowing downstream (Figure 3). In these rivers, small-scale irrigation operators at the upper stream over-extract the water and affect downstream users.

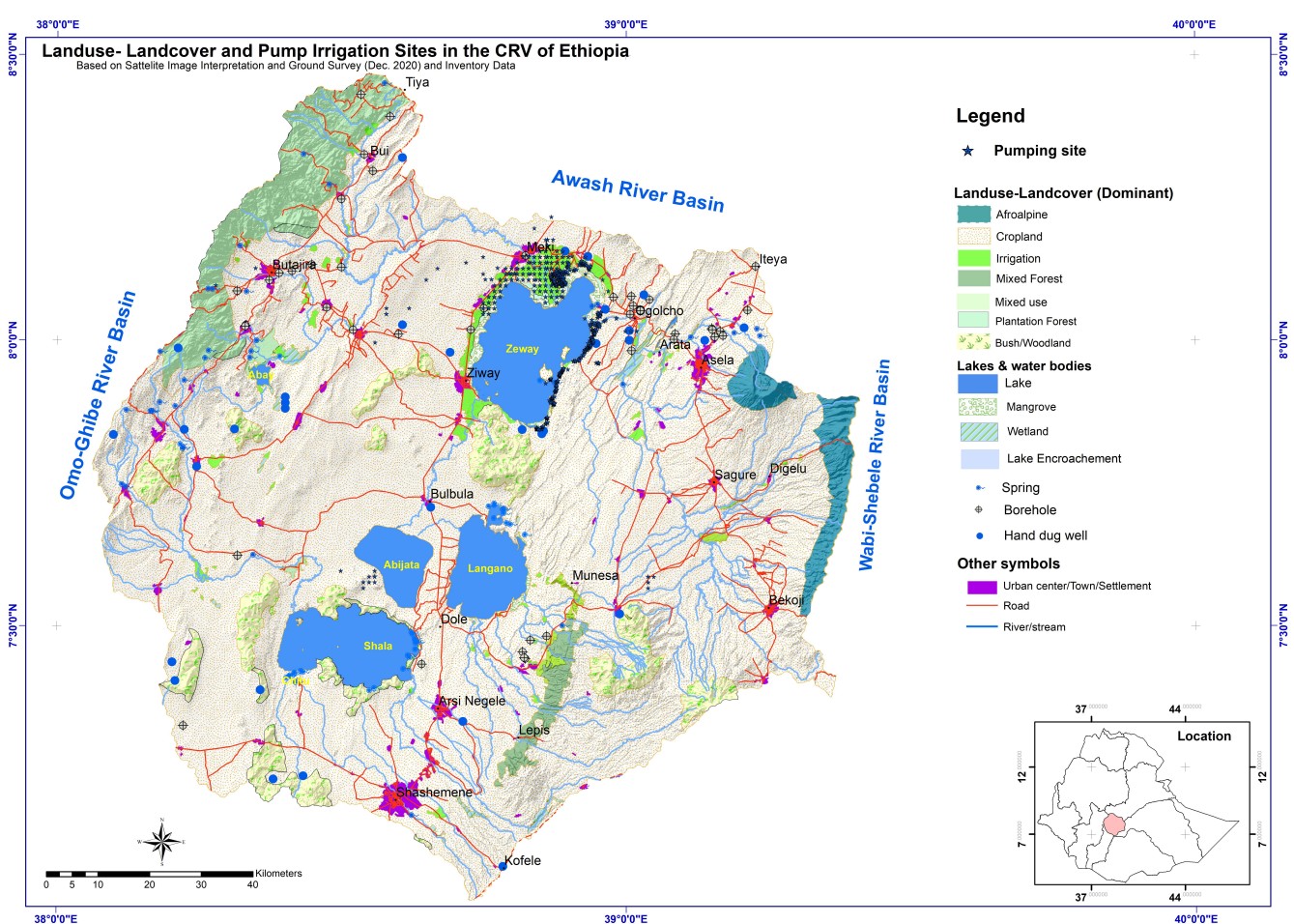

**Figure 3.** Water abstraction sites along rivers and around lakes in the upper CRV (note the distribution of pumping and diversion areas along the altitudinal gradients, showing the need for upstream-downstream negotiations).

The above factors lead to intense competition for water among the water users and types of water usage in the basin. Competitions lead to a mix of conflicting and cooperative interactions. The conflict mainly occurs when upstream users heavily consume water for irrigation, which seriously affects downstream users. An example is the dispute between the upstream and downstream water users in Meskan Woreda on the Yagulo River. The river has a low potential for irrigation during the dry season because it is mostly utilized by upstream water users. As a result, downstream farmers at the Agullo, Mori, and Jollie 2 and 3 kebeles use streams and underground water for their irrigation during the dry seasons since the volume of the Yagullo River decreases drastically. In this competitive water use environment, a clear allocation plan is unavailable; there is also a lack of effective governance of the water resource management, while the amount of water to be used every month and season is not fixed/planned according to the potential availability of water. Furthermore, given the intensified economic drive of the political-economic policy of the country, with the exception of domestic water supply, water use priority is given to investors. In addition, and emanating from poor investment in water supply, water insecurity is evident in many parts of the CRV where essential access to domestic water in the drier and lower-lying part of the sub-basin is unmet (see, for example, Figure 4 for a queue to obtain water for domestic use). However, different actors have cooperated to invest in and build communal water points (such as community ponds, deep underground water points, etc.) when the community is faced with acute water shortages, especially during some prolonged drought periods.

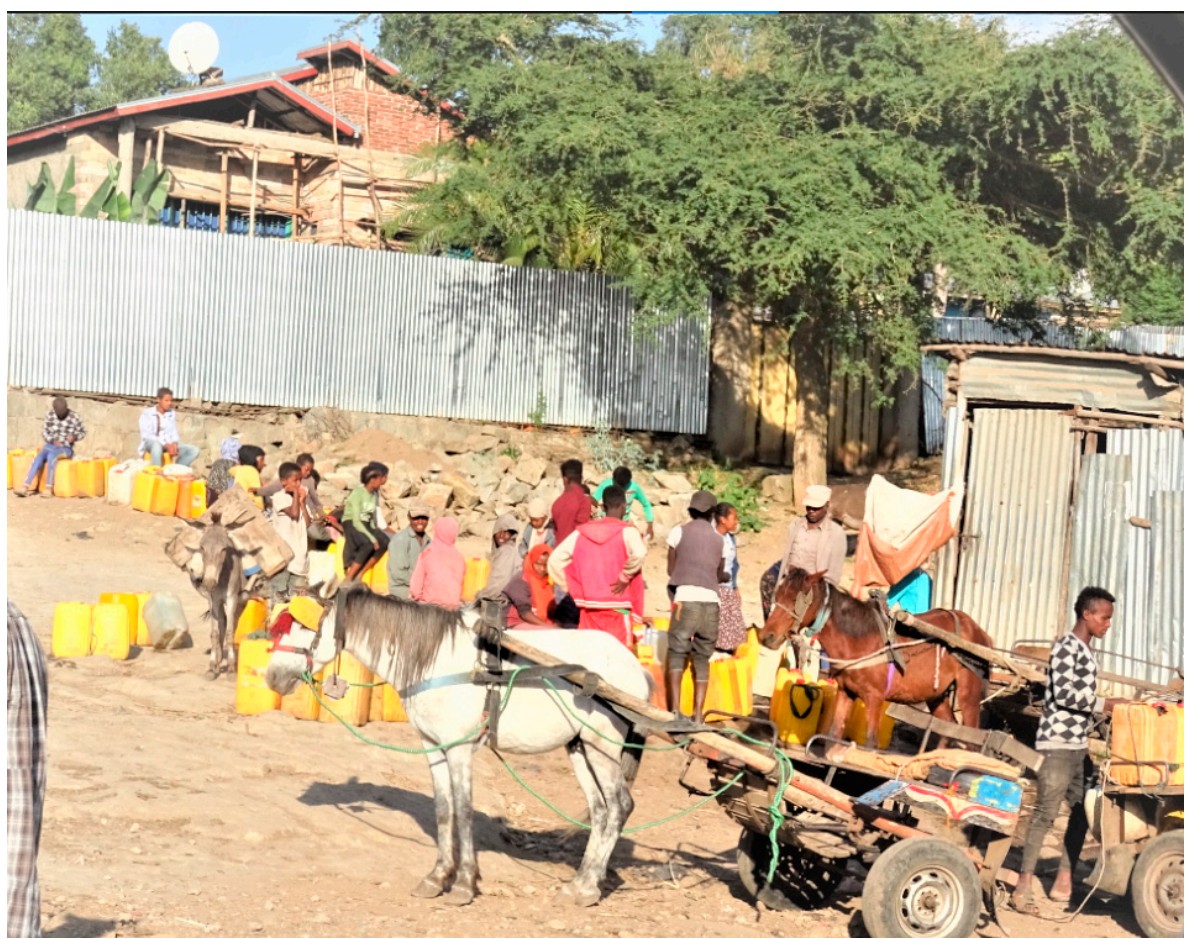

**Figure 4.** Rural water supply in Ziway Dugda woreda: photograph showing a line of yellow plastic jerrycans for domestic water, transported by a donkey cart from remote sources (photo by Taye Alemayehu, 2020).

### *3.3. The Discourses and Narratives on the Ground*

3.3.1. The Decentralized Resource Development Discourse/Narrative

One of the main discourses shaping water governance and policies in Ethiopia is the decentralization discourse. Agrawal and Ostrom observe that "decentralization has emerged as a major strategy for many nation-states to achieve development goals, provide public services, and undertake environmental conservation" [25] (p. 485). Proponents of decentralization argue that, unlike a centralized system, decentralized governance redistributes power, authority, resources, and accountability to lower levels of authority. It is a system of decision-making or a framework for participatory resource and political management at a regional level of administration [26–28].

In Ethiopia, decentralized resource governance and development formally appeared in the post-1991 government with a set of policies that comprehend fiscal, political, and administrative changes following the 1995 Constitution [29]. The ideological justifications for choosing decentralization as a governance structure by the then Ethiopian People's Revolutionary Front (EPRDF)-led government in 1991 include: a strong desire to achieve enhanced public participation through decentralized governance to break the poverty cycle in the country; empowering local communities; achieving consensual decision-making, equity, representation, accountability, and responsiveness; and serving as a means of preventing and containing ethnic conflicts and accommodating diversity [26,28,30].

The 1995 Constitution is the bedrock for the new federal government arrangement and the process of decentralized natural resource management. Among the major provisions of the Constitution pertaining to decentralized natural resource management are: Article

40/3, which states "the right to ownership of rural and urban land, as well as of all natural resources, is exclusively vested in the State and in the peoples of Ethiopia. . . . "; Article 51/11, on the powers and functions of the Federal Government regarding water resource management, stipulates "it shall determine and administer the utilization of the waters or rivers and lakes linking two or more States or crossing the boundaries of the national territorial jurisdiction"; and Article 52/2, which gives Regional States the power to "administer land and other natural resources in accordance with Federal laws". To ensure these provisions, particularly for managing water resources, the federal government and regional states established the Ministry of Water and Bureaus of Water, respectively. Furthermore, in accordance with the Constitution, the Federal Government issued a Water Resources Management Policy and Strategy in 1999 and 2000, respectively [31] and several related laws (Proclamation no. 197/2000 on water resource management; Regulation No. 117/2005 on water resource utilization including permit and fee; Proclamation No. 534/2007 on the establishment of the River Basin High Council and River Basin Authorities; Regulation No. 253/2011 on the establishment of the Rift Valley Lakes Basin High Council and Authority; and Regulation No. 441/2018 on the establishment of the Basin Development Authority), while regional states set out the duties and responsibilities of the water bureaus.

Researchers [6,7,32,33], however, argue that although reforms for decentralization aim to increase the effectiveness of water sector activities, the government would not keep its promises when confronted with political, socio-economic, and legal contexts from different actors in the country. An analysis of the legal framework concerning water resources use and management reveals that existing laws lack sufficient clarity in terms of assigning mandates and responsibilities based on the administrative regional states and river (lakes) basins. The problem stems from interpreting the constitutional decree of Articles 51/11 and 52/2d, which place jurisdictional boundaries on the responsibility to administer and utilize the waters of rivers and lakes. Explicitly stated, Article 51/11 gives responsibility to the federal government for managing and administering water resources that link two or more regions and transboundary water resources (lakes and rivers), while Article 52/2d makes regional states responsible for water resources that are found within the geographical jurisdiction of those states. However, because water resources are functionally and physically linked, any activity upstream affects the downstream water resources (including the runoffs or tributary stream). Hence, dividing the water management activities using political/administrative boundaries, in general, is contrary to the principles of IWRM and basin development, both of which are pillars of Ethiopian water resource policy. Furthermore, the subsequent laws enacted by the federal legislature and council of ministers have shown that mandates overlap the laws of regional states (e.g., Proclamation no. 197/2000 and Regulation no. 117/2005). One area of mandate overlaps concerns water permits. In this regard, the focus group discussion (FGD) and survey data of this study reveal that both the Rift Valley Lakes Basin Office and the two regions' water bureaus are granting water use permits for different sectors. For example, for large-scale irrigation (>3000 ha) and industries, the Basin Office and the Regional Government Investment Offices and other sector offices are involved in giving permits, while for small-scale irrigation (<200 ha) in both the Oromia and SNNP regions, zonal and woreda administrations and allied sector offices are granting permits (Key Informant Interview with the then Rift Valley Basin Development Authority and survey results, 2021). Hence, investors, developers, or anyone requesting permits must first find out which of the various departments will grant them a permit, which might prove difficult and could hinder transparency and accountability between those departments.

The effectiveness of decentralization can also be judged from the capacities of water management offices in their respective administrative tiers (in terms of budgetary allocation and management, technologies used, managerial capacities, operations, and maintenance of water infrastructure), where they are to be found ill-equipped in all aspects. Furthermore, field observations and the results of previous studies have identified serious weaknesses in these capacities [32].

3.3.2. Water-Centred Development Discourse and Narrative

Another key discourse is associated with the grand narrative that Ethiopia has rich water resources. This narrative encourages putting water at the heart of development policies and economic modernization plans. Several researchers contend that although Ethiopia is rich in water resources (both blue and green), endowed with fertile soil, and has huge areas of agricultural land, it does not meet its food security [34–36]. One plausible reason for Ethiopia's failure to achieve food security is the inefficient and insufficient utilization of its natural resources, particularly water resources. Since the 1950s, Ethiopia has had a vigorous interest in developing its water resources, as clearly stated in the three five-year economic development plans from 1957–1973 (during the Imperial period), the ten-year Perspective Plan of the period from 1984 to 1994 (during the socialist *Derg* period), the successive annual and five-year Growth and Transformation Plans from 1995–2021, and now the ten-year Perspective Development Plan for the period 2021–2030.

In these plans, the progression of water resources for agriculture, fisheries, energy generation, livestock, tourism, and enhancing the domestic water supply were aimed at propelling the growth of the country and advocating water-centered development. In this regard, planned modern irrigation agriculture in the CRV started in the 1950s during the Imperial regime. Regarding overall CRV development, the Third Five-Year Development Plan (1968) remarks the following:

"Because of fairly rapid recent development in the Rift Valley lakes a systematic inventory of the land and water resources of the area will be undertaken early in the plan period to enable national land use planning. Such planning is urgently needed, in particular as it appears that some land uses may be competitive, such as lakeshore tourism, lakeshore agriculture, and the conservation of wildlife" [37] (pp. 113–114).

The 1984–1994 Ten-Year National Plan made clear its objective to transform the economy, along socialist lines of development: "The plan has set itself the lofty and difficult task of propelling Ethiopia out of the abyss of economic backwardness by enunciating appropriate development objectives and by creating favourable conditions for their realization" [38] (p. 14), and "its broad goals are the structural transformation of the economy through the development of the country's productive forces and raising the living standard of the population" (p. 18). In the water sector, the plan was to use medium- and large-scale irrigation projects to develop 126,000 hectares of land, establish 1900 meteorological and 557 hydrological stations for improving the monitoring and investigation of the country's hydrological resources, and reach 13 million rural and 7 million urban inhabitants with water supply (altogether 47.6% of the total population at that time). The newly approved Ten-Year Development Plan (2021–2030) has a renewed commitment to use water as the major natural resource for national development.

Awulcahew argues that the development of water resources is a critical process that could enable the country to move higher up its development ladder. He further asserts that, if successful, irrigation in Ethiopia could sustain the agricultural development of the country by contributing up to ETB 140 billion to the economy and potentially ensuring the food security of 6 million households. For the period from 2002 to 2021, several five-year plans envisaged for the rapid growth of irrigated land were devised. As for the energy sector, the country's source of electricity derives mainly from hydropower through the construction of hydroelectric dams [36]. In general, water development was considered a strategic resource for the development of the country. However, access and benefits from water resource development still failed to reach a considerable part of the population. In the CRV, irrigation agriculture was mainly in place to produce high-value horticultural crops for industry, for export, and, to some extent, for urban consumers.

The discourse of water-centered development is mainly prompted by the alarming national food insecurity and the food demand of an increasing population. The major argument here is that the growing food demand cannot be met by conventional rain-fed agriculture alone. Furthermore, there is a growing demand for agricultural products to supply raw materials for agro-industries and for export to obtain foreign earnings. The

hydropower needed to propel the economy is also given due attention. This discourse has implications for water security in two opposing respects. The first implication is that it imposes high pressure on the water resource by expanding irrigated agriculture to a degree that negatively affects water resources. The second implication is that it largely contributes to ensuring water security by meeting the growing demand for water and reducing the competition over it. This means it paves the way for enhanced utilization of the untapped water resource potential of the country, or, contrarily, it may contribute to water insecurity for some water users. Thus, to reduce the negative trade-offs of a water-centered development plan, appropriate water-conserving technologies and appropriate methods of utilization should be considered when forming that plan.

### 3.3.3. Market-Led Natural Resource/Water Resource Development Discourse/Narrative

Being well aware of the importance of agriculture, the government of Ethiopia has been implementing an Agricultural Development-Led Industrialization (ADLI) strategy since 1994, which makes agriculture the engine of other sector developments. The aims of the strategy are to improve the coverage and quality of agricultural extension services; promote better and more efficient use of land and water resources; enhance access to financial services, particularly for low-income citizens, including women and youth who have cooperative and financial plans; improve access to domestic and export markets; and provide rural infrastructure [39].

While putting ADLI into practice, the government has prepared and implemented several strategies and Five-Year Development Plans since 2001. These development strategies and policy frameworks have led to the implementation of the Sustainable Development and Poverty Reduction Program (SDPRP) from 2001 to 2005, the Program for Accelerated and Sustained Development to End Poverty (PASDEP) from 2006 to 2010, and the Growth and Transformation Plan (GTP) I (2010–2015) and GTP II (2016–2020) [40]. In all these strategies and plans, water sector development is emphasized and positioned as an engine of development.

Based on the different policy frameworks and implementation strategies, the government has encouraged export-oriented horticultural development, including floriculture, which essentially depends on intensive irrigation. The floriculture industry has grown from around 72 flower farms being active in 2009 [41] to 126 in 2018 (Horticulture Producer Exporters Association (EHPEA): http://www.ehpea.org (accessed on 5 June 2022) (see also Figures S2, S4 and S5 on the expansion of horticulture farm close to Lake Ziway and irrigated wheat farm in the middle altitude). The expansion of the floriculture industry and other export-oriented agricultural practices in the CRV has its own influence on the water use and governance system since it increases the number of water users and actors involved in the basin. When there are more actors and water users, there are often more (mostly incompatible) interests and positions to start water competitions and conflicts.

In general, heavy water use as a means of production in such industries and in other types of economic activity in the CRV basin, including both large- and small-scale activities, aggravates existing water shortages and competition among actors. Water is considered an economic good or commodity because of its economic value. As a result, investments in water infrastructure, such as the construction of community dams and irrigation schemes, can act as a stimulus for local and regional development efforts. However, when there are diverse and sometimes incompatible water uses and users, it is imperative to follow the principle of giving priority to water use for economic development.

In conclusion, the urge to transition from subsistence-based to market-based natural resource development in general and water resource development in particular has its own effect on water governance. In the CRV basin, water resources are under increasing pressure due to competition and divergence among users, as well as climate change. Water for domestic use, livestock, fisheries, industries (floriculture, soda ash, breweries, etc.), irrigation practices, and the environment are some of the competing water uses and users in the CRV basin that have a bearing on increasing water shortages, competition, and conflicts.

3.3.4. Land/Water Resources Degradation Narrative

(i). Upland degradation discourse as an immediate cause of water degradation

Land degradation has become a prominent theme of policy discourse in Ethiopia since the 1970s. Since the mid-1970s, the problem of land degradation has been portrayed as a major threat to the country's survival. Since then, the Ethiopian Government and many stakeholders (such as the World Food Program in its Food for Work program, the World Bank, civil societies, and several donors) have embarked on interventions of different scales to reverse land degradation through land rehabilitation and conservation campaigns. Such discourse holders have argued and identified various causes and consequences of land degradation. Concerning the causes of land degradation, the discourse holders pinpointed a range of reasons: some believe that land degradation is the result of population increase and due to traditional land use systems; some argue that it is due to the unfair distribution of land for agriculture (the narratives in the 1970s and 1980s were due to landlordism, the eviction of indigenous people from fertile agricultural land to marginalized land); some argue that the major bottleneck is the absence of an effective land use policy and land use plan, etc. In general, each of the explanations above on the causes of land degradation and the recommended solutions were the results of political and ecological thoughts of different periods.

As briefly described in Section 2.2, the CRV has diverse geo-ecology (diverse altitudinal and agro-climatic belts, distinct ecosystems, and rich biodiversity). Though this ecosystem is very important for its diversity, the basin is currently among the hotspot areas of soil, water, and biodiversity degradation in the country. Until the 1970s, this area was sparsely populated, and accordingly, there was little pressure on the ecosystems. For instance, a study by Meshesha et al. revealed that in 1973, dense acacia woodland and forestland covered about 44% of the area. In 2006, the forestland had diminished by 66.3% and the woodland by 69.2% [42]. Several other studies covering the whole CRV or parts of the sub-basin reveal rapid land cover changes in the last five decades [43–45]. The major conversion of this land cover to cropland was undertaken by smallholder farmers and commercial agriculture through the introduction of large-scale livestock ranches, tourist lodges, and the expansion of urban settlements. Furthermore, the encroachment of cultivated lands into the montane and mid-altitude forests reached its highest level in the last three decades. These conversions become major causes of excessive soil erosion, flooding, and sedimentation at the Ziway, Langano, and Abijata lakes. Deforestation of ridges and water divide areas has particularly contributed to the formation of badlands and gullies of varying severity (see Figure S2).

In addition to Land Use and Land Cover (LULC), several studies have revealed the sereneness of soil erosion and sediment yield, and climate change has affected the water resources of the basin in general and the lakes in particular [18,46–52]. In addition, overgrazing and trampling due to a large livestock population have contributed to soil degradation (changes in soil bulk density, soil structure, soil porosity, etc.). Soil degradation (mainly soil erosion and sediment yield) from deforested steep slopes has been increasing over the years and has negatively affected the lakes in different ways [42,46,51–54]. Notably, the hydrological effects of this rapid land conversion and the resultant land degradation are observed in the reduction of lake volumes. For example, Aga et al. (2019) modeled the amount of sediment deposition in the bed of Lake Ziway to be 2.039 million tons every year, which would contribute to the loss of the lake's water volume by 0.106% annually. Another study by Gadissa et al. [55] concluded that the average annual sediment yield entering Lake Ziway was 431.05 tons/km$^2$ and 322.82 tons/km$^2$ for the Meki and Katar rivers, respectively.

In their historical area coverage assessment of Lake Abijata, Temesgen and his colleagues [56] reported a 5.6% reduction of the lake's area in 1986 from the 1973 level, with no change until 2000 and then a 46% reduction in the period between 2000 and 2006; Wagaw, Mengistu, and Getahun [57] also revealed the lake's retreat from 215 km$^2$ in 1980 to 87 km$^2$ in 2016, which in terms of water volume is a reduction from 1605 MCM to

less than 400 MCM. One frequently mentioned reason for the shrinkage of the lake is the establishment of the Abijata Soda Ash Factory.

Our several interview results show that the most frequently mentioned reason is usually given by local residents (particularly the youth), who assert that the lake's water is "their water resource", and some experts agree that the lake's volume has dwindled because of the Soda Ash Factory. The Soda Ash Factory was established at the shores of Lake Abijata in 1989 and produces soda ash, a form of hydrous sodium carbonate ($Na_2CO_3$), by evaporating water pumped from the lake in artificially constructed ponds for the crystallization of trona. It has been reported that the factory was pumping about 1.5 million $m^3$ of water annually to produce about 10,000 tons of soda ash per year [58], which is <0.4% of the current lake volume. For several reasons, the factory halted its production for the last three years but now plans a resumption. Given the relatively small amount of annual abstraction and that the water is saline and never used for irrigation, domestic water supply, or animal watering, the reason given by local residents is doubtful. Their assertion is said to be motivated by politics, as the water has never been used by the residents, which is driven by the current political activism and also shows a sort of political power shift to the locals. On the contrary, livestock owners are happy with the extensive retreat of the lake, which yields large tracts of grazing land (Figure 5 shows a herd of livestock grazing on the shore of the lake; field visit, 2020). The FGD with Soda Ash Factory staff revealed that the shrinkage of the lake is due to smallholders and large-scale commercial farms over-abstracting water from Lake Ziway and the Bulbula river that drains into this terminal lake, heavy sedimentation of the lake from the upslope due to land cover changes, and climate change.

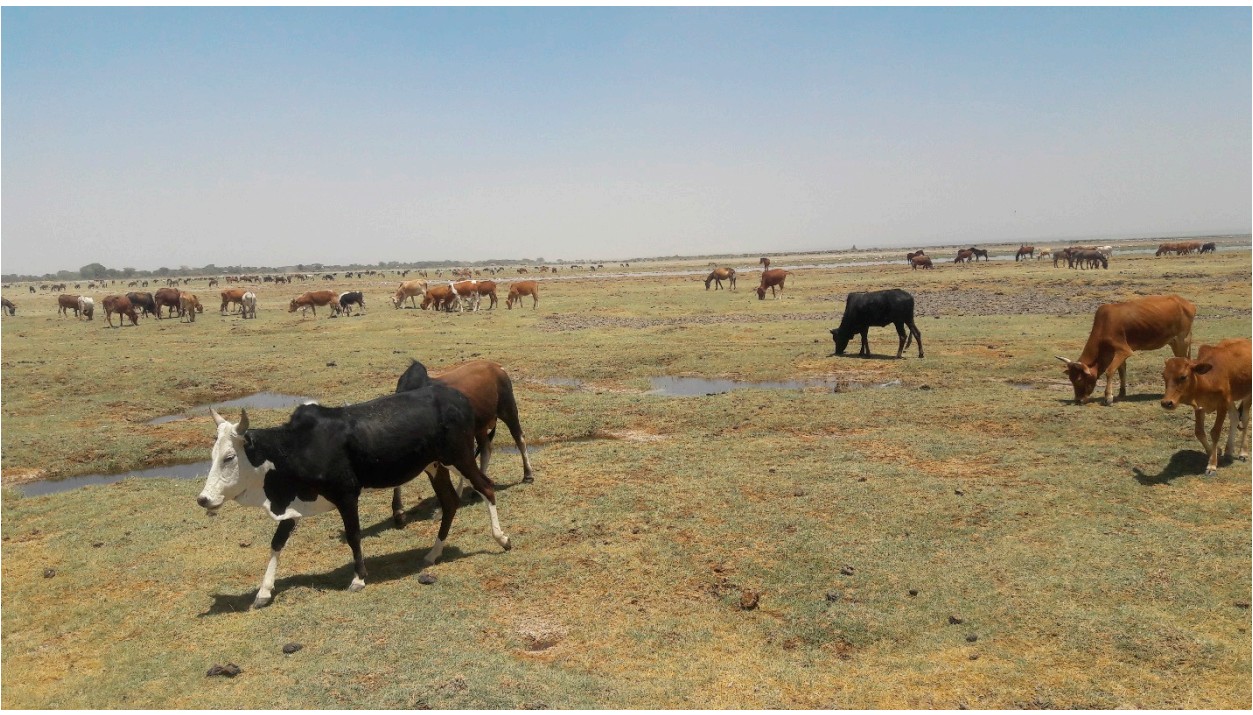

**Figure 5.** A herd of livestock grazing on an exposed part of the bed of Lake Abijata (note that the lake is saline) (Photo: Amare, 2020).

Expansion of grazing land by deforestation also causes soil erosion and sedimentation of the lakes. Overgrazing and large numbers of livestock cause the compaction and trampling of soils and result in low soil infiltration, high surface runoff/flooding, and topsoil erosion. The local community extracts salt soil from the shore to be sold as complementary feed for livestock, which, along with sand extraction (at the shore and in

the Acacia woodland), leads to deforestation and sediment movement to the lake. All the above factors contribute to hydrological regime disruption.

In conclusion, many scientific studies and other opinions from experts and policymakers, and the stories/beliefs of the local population, argue that the landscape has greatly changed and the trend of land degradation persists year after year. They further reiterate that reversing land degradation is an urgent action needed to achieve the desired sustainable development.

(ii).  Water pollution narrative

Water quality deterioration in the basin is a growing concern for many actors, particularly the local people, environmentalists, and local political activists. The study of Merga, Mengistie, Faber, and den Brink [59] in Lake Ziway showed that nutrients, pesticides, and trace metals had accumulated there at an increasing rate. A report by Jansen and Harmsen [60] also confirms the deterioration of water quality, particularly at the shores of Lake Ziway, where representative samples were mostly taken. Furthermore, the quality of groundwater samples was found to be "unsuitable for long term agricultural use due to their high salinity and sodium adsorption ratio, which has implications for soil permeability, as well as elevated bicarbonate, boron and residual sodium carbonate concentrations" [61]. During the FGD, the local people, as well as some experts, express their concerns for the health of the people and livestock using water from streams and lakes in the lowlands due to possible contamination by agrochemicals, either from smallholder farmers or from leakages and discharges of polluted water from commercial floriculture.

Some experts expressed their worries over the increasing use of pesticides by smallholder farmers with no knowledge of the correct dosage and ways of using them or their consequences for human and livestock health, the long-term health of their farmlands, and the environment at large. In this regard, Mengistie et al. argue that farmers use pesticides without considering safety recommendations: "they use unsafe storage facilities, ignore risks and safety instructions, do not use protective devices when applying pesticides, and dispose containers unsafely" [62] (p. 301). There is also a belief among many stakeholders that the floriculture companies are opaque in declaring the amount and type of agrochemicals they are using in their farms, and there is a general conviction that there is an overuse of pesticides to the detriment of all aspects of the community's health and environmental protection.

### 3.3.5. The Water Scarcity Narrative: The Aral Sea Syndrome in the Making

The German Environment Advisory Group in 1996 came up with the concept of Aral Sea Syndrome to depict unsustainable, uncoordinated, and over-utilized water resources for development derived from water bodies situated in arid and semi-arid environments, which eventually caused the drying up of such water bodies and the devastation of the ecosystem, an illustrative example being the rapid decline of the Aral Sea in Central Asia [63]. In the same line of argument, several researchers [64–66] labelled the CRV a freshwater-scarce sub-basin heavily affected by human activity that is threatened to collapse, despite the fact that it hosts several large and small lakes. This scarcity, they argue, is mainly due to a rapidly growing demand for and over-abstraction of water from the lakes, as well as from streams feeding the lakes, for various uses (domestic, irrigation, livestock, and industry). Other researchers attribute the water scarcity to climate change and variability and to upland degradation that contributed to the sedimentation of the lakes (see Section 3.3.4 above). They think that due to the rising demand, the water supply could dwindle faster and some lakes could dry up entirely.

Similarly, planners, environmentalists, and even the local community contend that the over-abstraction and uncoordinated utilization of Lake Ziway and Lake Abijata threaten the existence of these lakes. For instance, Lake Abijata's area decreased by 5.2% between 1973 and 2000 and by a further 46% between 2000 and 2006 (Temesgen et al., 2013). Likewise, the water level of Lake Ziway is falling because of increased irrigation for agriculture around the lake. Furthermore, over-abstraction is reported not only at Lake Ziway and

Lake Abijata but also at the Katar, Bulbula, and Meki rivers and the groundwater in many parts of the basin [58]. Goshime and his associates reached the conclusion that "the amount of water withdrawal from the lake [Ziway] for irrigation water use is 37 million $m^3$ per year. This led to 0.36 m drop in the lake level which corresponded to 18 $km^2$ reductions in the lake surface area. This consequently resulted in a reduction of mean annual lake volume by 162 million $m^3$ from 1986–2000, which accounts for 23% of the total lake inflow from rivers" [67] (p. 67).

The concerns about the scarcity of water are also expressed by livestock herders, smallholder irrigation practitioners, the Soda Ash Factory, Abijata-Shala National Park, and residents who require it for domestic use. The Soda Ash Factory experts and the Abijata-Shalla National Park, for example, complain that the over-abstraction of water from Lake Ziway affects the outflow to the Bulbula River and later reduces the inflow of water to Lake Abijata. Livestock herders identify an insufficiency of water for their livestock as their chief problem, especially during the months from December to May.

3.3.6. Weak Institutions and Weak Water Use System Narrative

A record of official water laws in the country can be traced as far back as the 15th century with the adoption of *Fetha Negest* ("Justice of the Kings"), which sets down the earliest formal rules that constitute both spiritual (related to the faith of the Orthodox Church) and secular laws, of which the water issue is one. It was introduced in Ethiopia during the reign of "King of Kings" Zar'a Yaqob (1434–1468) [68]. However, in modern times, water institutions are descended from the attempt to institutionalize municipal water management in the 1940s and, more precisely, the adoption of the second written Constitution of the country ratified in 1955 [69]. Notwithstanding the long history of water institutions, they are today heavily criticized for their weak performance on the ground, for inadequately meeting the interests of those with growing demand for water resources, and for failing to take full advantage of the resource potential. This critique emerged from different perspectives. First, institutions in general and water resource governance in particular have frequently changed in the last six decades (see Appendix B). Second, there has been weak institutional coordination among different sectors directly or indirectly involved in the management of water resources [70]. Third, there is a lack of organizational capacity for water governance at all levels, from the local to the federal [32,71]. Fourth, there is a dearth of reliable long-term data on hydrometeorology and hydrology, as well as on the potential of the basin's water resources, as required for better planning. Fifth, the legal framework in which to manage the water resources lacks clarity, with several duplicated roles between different sector organizations and an absence of clear demarcation and mandates to regulate, allocate, protect, and develop the water resources for both rural and urban users and various sectors. Sixth, the failure to prevent or mediate conflicts of different natures at different scales, including between the upstream and downstream users, where in this case there is a need to not only manage water resources but also protect the highland ecosystems and landscapes from degradation as they are primary sources of water (see Section 3.3.4 above).

The following points are presented in support of the argument for the weakness of institutions working on water:

First, researchers argue that institutional coordination is lacking for the effective implementation of water resources development, management plans, and IWRM [6,23]. With reference to the spatial scale and administrative power relations, institutional interplay in the CRV follows two distinct principles: hydrologic and political administrative boundaries. Institutions under these two structures can also have horizontal and vertical interactions. Powers and responsibilities from both directions converge upwards until they intersect at the level of the MoWE, a national apex body for the administration of water resources. With regard to arrangements based on hydrological boundaries, the river basin development authority (RBDA) at a national level and the Rift Valley Basin Development Office (RVBDO) form the main structure, while the Regional Water Bureaus (RWBs) and the

Zonal and woreda water offices form the lower parts of the structure according to the political-administrative system. These two power structures, according to the RVBDO experts, are, however, poorly integrated to implement IWRM. One vivid example rests on the issuance of permits for water resource use in the agriculture industry. While the proclamation no. mandated the basin authority to issue licenses and permits, the Regional Bureaus of Water also have that authority vested in them through the constitution. Hence, there is a duplication of effort in this regard. Financing is another issue. Our fieldwork confirmed that the sector at regional, zonal, and woreda level are poorly financed. The RBO is basically financing a few works on watershed management to protect water bodies, which are far from adequate, and on planning activities.

Second, irrigation methods used by smallholder farmers are inefficient; they almost always use a furrow irrigation system, which is extremely wasteful [72].

Third, interview results with Adami Tulu, East Meskan, and Ziway Dugda, the Woredas Department of Agriculture, and the field observations reveal that there is no regulation on where to use groundwater and how much water is to be abstracted from rivers. These farmers are usually contract farmers motivated more by profit than sustainable water resource use. There is no or very little effort to reduce soil erosion and sedimentation at the water bodies, both rivers and lakes, and there is no effective regulation to protect the water bodies from pollution from point and non-point pollution sources. Major polluters are those involved in commercial floriculture and smallholder horticultural farms that use pesticides and weedicides. There is the contention by experts that the floriculture companies have a powerful say because they are privileged by the government because they are generating foreign currency. The woreda-level experts have neither the trained human power nor the institutional capacity to regulate and monitor non-point source polluters. Fourth, there is no or very little work conducted to improve water storage during the rainy months, from the small household scale to the community level and to the large-scale reservoir of water, in order to enhance the water availability element of water security.

Fifth, there is no clear and effective regulatory framework for mediating between upstream and downstream users and addressing the needs of the disadvantaged. In the policy, these things are addressed; nevertheless, their implementation at the grassroots is minimal.

## 4. Discussion: The Discursive Impacts of the Narratives in Water Governance in the CRV

Intricate relationships between discourses offer explanations for the water security issues in the CRV (Figure 6). In line with decentralized water resource management in the CRV (decentralization discourse), bureaus and offices of water resource management have been established in the regional states following the administrative tiers of the country. In parallel, the Rift Valley Basin Development Office adopted the hydrological boundary (basin development) approach as its preferred approach to basin development. These two systems of water resource management, however, were found to be incompatible with effective water resource management on several occasions. The limitations include mandate overlap and inconsistency in priorities. In this regard, researchers [28,73] contend that although decentralization is constitutionally guaranteed in Ethiopia, efficient and effective decentralization has neither been fully implemented nor brought better results than the centralized natural resources administration system. As discussed earlier, the existing laws lack sufficient clarity in terms of assigning mandates and responsibilities as appropriate to administrative regional states and river (lake) basins. Furthermore, in this decentralized political governance system, power and responsibilities were envisioned to devolve from the federal government down to the regional, zonal, and woreda levels; however, weak institutional capacity at the lower administrative levels (in terms of trained human capacity and budgeting) has greatly impeded its success in delivering the expected good resource governance [74].

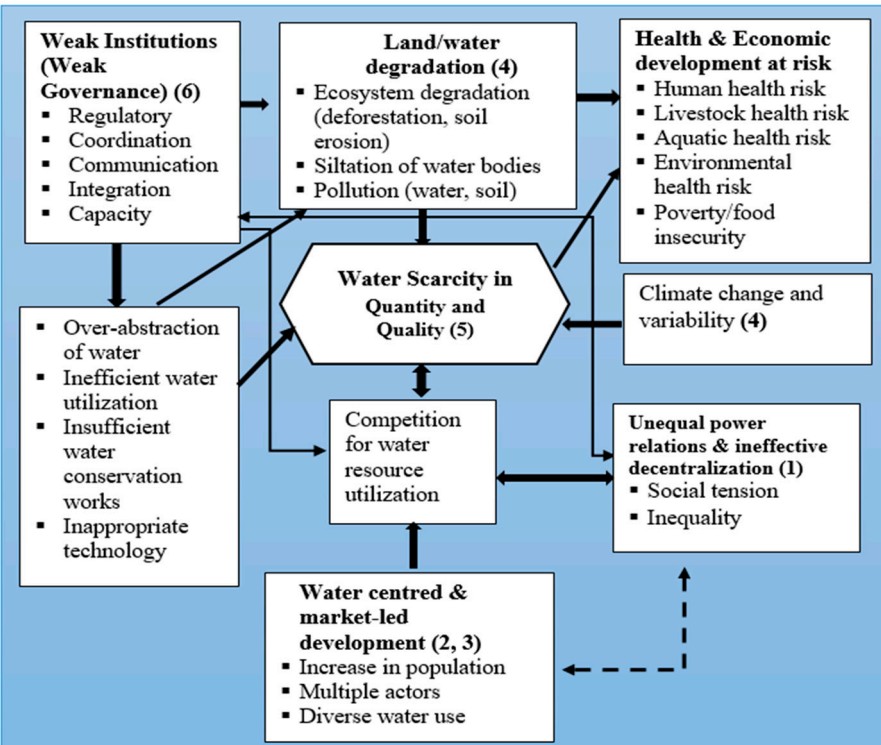

**Figure 6.** Interrelationships between and reinforced feedback on the dominant water resource management discourses and narratives in the CRV sub-basin. Institutions and governance are linked to all kinds of issues. Note that with the identified problems, the consequences of these interlinked problems would result in water and food insecurity. On the other hand, many of the problems could be solved by improving water and environmental governance. The dotted line indicates an indirect relationship. Numbers in the bracket indicate the major discourses and narratives discussed in detail in the Section 3 (also linked to Figure 2).

The growing competing claims for water resources by multiple water users and the resultant conflicts between the different types of actors are mainly rooted in the existence of weak institutions and poor water governance and development in the CRV basin. To illustrate this claim, the sub-basin's demand for water supply for domestic use has never been met; the heavy abstraction of irrigation water in upstream areas (e.g., in Katar, Bulbula, Yagullo, Hulluka, and Meki rivers) usually leaves little for downstream users; irrigation water for dry-season wheat production is given priority by the government over vegetable crops against the preferences of farmers; and commercial flower farms are given disproportionate government support to produce these products for the export market than smallholder farmers. The idea of water governance is fundamentally about tackling such competitive or potentially competitive situations where two or more actors/parties seek access to the same water resource [75]. The existence of poor governance usually emanates from the performance of weak institutions and unfulfilled political commitments. The competitive interests among the different actors and water users can be dealt with by establishing well-designed and strong water institutions capable of realizing efficient and effective distribution and regulation mechanisms for access to, control over, and management of water resources.

The existence of weak formal institutions in general leads to environmental degradation, improper water management (including uncontrolled water abstraction), water pollution, and degradation of upland (water source) areas. Weak formal institutions would also have little capacity for managing the equity of resource distribution and would be unable to manage multiple interests and emerging conflicts. It is evident from the field assessments that customary local-level institutions are much better and more effective at

tackling local-level conflicts arising from water resource usage. The building blocks of these customary institutions are the societies' cultural elements, such as norms, values, sanctions, principles, taboos, and rules that govern the access, control, use, and distribution of water resources. Therefore, in order to develop a sustainable system of water resource governance in an increasingly degraded environment and vulnerable climatic conditions, there is a need to establish robust institutional frameworks. There are also calls for collaboratory, customary, and statutory institutions to produce a sustainable system of water resource governance.

The water-cantered and market-led water resource development discourses obtained renewed interest, which led to the expansion of commercial irrigation farms for wheat and horticultural crops and the expansion of agro-industries through the establishment of agro-industrial parks. All these will abstract huge amounts of water from both surface and groundwater for different uses. This recent government policy exacerbates competition over water resources. These two discourses, which are backed by the regional and federal governments, are exacerbating the claim of the water scarcity narrative (a narrative shared by local residents and researchers) and the claim regarding the burden on environmental flow (a voice mainly from researchers and environmentalists).

Furthermore, the increasing trends of land degradation, siltation of lakes, abstraction of water, inefficient water utilization, and weak water management are related to land/water degradation discourses. Coupled with climate change and increased water abstraction, all sorts of degradation discourses mentioned above have aggravated the severity of water scarcity problems. The increasing competition for water resources and resulting conflicts have a direct bearing on social tensions and animosities that heighten polarized and asymmetrical power relations among the various actors. The competition would further lead to inequitable and inefficient water use systems. Extensive water pollution caused by point and non-point pollution, such as discharges of untreated water from several flower farms, agro-industries, factories, and elsewhere, and the heavy application of pesticides and weedicides in commercial farms, pose a range of threats to human and animal health and the wider environment. There are also severe gully development and soil erosion challenges in the CRV basin as a result of long-term human activities. Anthropogenic forces that alter the physical landscape through environmentally unfriendly infrastructural development also cause considerable soil erosion, which has an adverse effect on surface water bodies. Accordingly, sediment control is an important consideration for catchment management planning in the CRV basin. Otherwise, the sedimentation of lakes and other water bodies and water pollution in the area will become worse than ever before.

Furthermore, water governance involves the procedures through which decisions are made. The decisions made by different actors are greatly influenced by the existence of power asymmetry among the various actors. The presence of such power asymmetries among conflicting parties affects the fairness of decisions on water distribution. This calls for developing governance mechanisms for just, equitable, and sustainable water resource use and management that would fulfil the water security needs of all.

There are concerns regarding the power relations and intensity of water conflicts in the CRV basin among different actors. For example, (a) power relations between the upstream and downstream water users increasingly became competitive and unfriendly, a situation that emanates from locational advantages and disadvantages for the upstream and downstream users. The upstream users exercise more power over those downstream users in abstracting water for irrigation without due consideration of the needs of downstream users. In this regard, there is no law mediating a water-sharing agreement between upstream and downstream users; (b) the other power emanates from the "rights" bestowed on the local residents through the principle of decentralization, where users who do not belong to the community are considered outsiders. This notion consecutively led to violations of the rights of those groups to access water. For example, the Soda Ash Factory at the shore of Lake Abaya was considered an outsider, and it was threatened by the local youth to stop production. A sense of "localism" is now observed in many places in the

basin; (c) another power relation in water resource use is observed by being a member and non-member of the irrigation water user association, where the latter only receives the right to use water for irrigation based on the goodwill of the members; (d) the immense support that large commercial farms and agro-industries have received water use from the highest level of government (federal and regional) makes them highly powerful than woreda level regulatory institutions, thereby the latter unable to effectively monitor the water use. Such power relations and competitions over water use lead to the emergence of diverse socio-political voices and contestations over the access and use of water resources through the different above-discussed discourses and narratives at various levels, from the community to the central government. A "battle" of discourses and narratives reflects the actors' power dynamics and their struggle to influence water governance in the CRV and in the broader socio-political context of Ethiopia. This line of thought, however, merits further research.

Furthermore, there is an unclear distribution of powers, duties, responsibilities, and rights levied on the different players in the relations between policymakers and/or decision-makers. The confusion emanates from the Constitution Articles 51/5, 52/2d, 40/3, and 92, which are relevant articles on land and natural resource management but not explicit on the power share as it concerns natural resource management. For instance, the Constitution indicates that the regional states shall have the right to manage natural resources, but for those shared by more than one state, the federal government will act. However, this does not indicate where exactly the trans-regional water flow starts because the transboundary streams first originate as small streams and tributaries in the uplands. In fact, in general, water sources are upland, and so the conservation of these areas is necessary. Any degradation in these uplands has serious consequences for the water flow downstream. The same issue can be raised with respect to groundwater, whose flow directions may not coincide with the surface watersheds. In general, since surface water flow starts from the upland and since lakes or major streams are functionally linked between the upland and valleys, administrative boundaries are not conducive to managing resources—particularly water at the watershed or in the basin—in a holistic manner. Healthy uplands are required because the water is fed by the uplands. Here there is confusion over who would have the ultimate right to water use when different users appear at one location: who gets first use and at what quantity? Regarding the different uses of water, Ethiopian Water Resources Management Proclamation no. 197/2000 Article 7(1), makes clear that domestic use takes precedence over all other uses. However, there is no metering on the rivers, and there is no restriction on the amount of extraction for other uses, and as a result, some rivers have dried up, causing a problem for residents.

Land/water degradation can be manifested as the long-term reduction or loss of at least one of the following conditions: biological productivity, ecological integrity, or value to humans. The major causes of land degradation in the CRV sub-basin are land use changes and unsustainable land management. These causes are considered direct human causes of land degradation, with agriculture the dominant sector due to its conversion of woody vegetation land into agricultural land, as well as low soil and water conservation practices. Meanwhile, a shift from pastoral and agro-pastoral production systems to sedentary ways of life, particularly in the lowland parts of the CRV basin, has also brought land degradation. Land use changes should be considered where current agricultural patterns are no longer sustainable in terms of water consumption. Land/water degradation affects human health, livestock health, aquatic life, and the ecosystem altogether in multiple ways by interacting with social, political, cultural, and economic elements, including markets, technology, inequality, and demographic change. The prevalence of land degradation, the expansion of desertification, and reoccurring droughts have negative effects on the availability, quantity, and quality of water resources, which results in water scarcity.

The degradation of ecosystems in this basin due to the unwise use and overexploitation of resources with the intention to achieve short-term economic goals has had direct medium- and long-term negative effects on social welfare. The cause of ecosystem degradation and

loss is often due to a failure to appreciate the full value of the functions provided by such systems. Therefore, in order to ensure the economic, environmental, and social benefits of sustainable development in the region, there should be a concerted, integral community development effort. This can be mediated by robust institutions not in isolation but holistically from a systemic integrative perspective.

## 5. Conclusions

This paper presented several interlinked discourses, narratives, and debates on water resource management in the context of overall natural resource management and its implications for water governance in the CRV basin. The discourses are: decentralization, water-centred development, market-led natural/water resource development, land/water resource degradation, water scarcity, and weak water resource governance. The narrative/discourse-holders range from local residents to international actors with diverse interests and powers. These actors, by way of their discourses and voices, have the potential to influence policymaking. The analyses of each of the discourses and narratives are strongly interlinked with each other as causes and effects. Some of the discourses have conflicting perspectives or priorities. Few of the discourses emanate from speculative data and information, while some are driven by political orientations such as "localism". In conclusion, the presence of varied discourses and narratives implies the need to understand water resource development from multiple perspectives (resource base/endowment, socio-political arena, equity, market, and capacity) and from varied interests that need to implement a systems approach in the attempt to resolve the issue of water governance. A "proliferation" of competing discourses and narratives reflects the actors' power dynamics and their struggle to influence water governance in the CRV and in the broader socio-political context of Ethiopia. This line of thought, however, requires further research.

The governance of natural resources in a wider context is an important consideration to improve the water security of the sub-basin amid alarming climate change, a growing population, and the rise in water demand from different sectors. Strengthening the capacities of institutions for resource governance in terms of budget, human resources, operations, and maintenance capacities is essential, as are clear and explicit laws, when seeking to improve water governance for sustainable development.

## 6. End Note

1. *Derg*—is the provisional Military Council that deposed the Emperor Hailieslassies's regime in 1974 through the popular revolution and changed the imperial government's ideology to socialist ideology.

2. In Ethiopia, woreda (equivalent to district) is the second-smallest administrative unit, the smallest being *kebele* (a neighbourhood of several villages).

3. Ethiopia has 12 major river/lake basins, the Rift Valley Lakes Basin being the one at which the CRV is situated. There are several estimates of the water potential of the country. Often-cited estimates include about 122 billion cubic meters (BCM) of annual runoff [36] and about 26 [76] to 47 BCM [77] of groundwater. However, due to the lack of adequate water storage infrastructure and the large spatial and temporal variations in rainfall patterns, there is not enough water for most users to fulfill their demands [36]. It is also often argued that, although the country is endowed with substantial irrigation and hydropower potential, the harnessed amount is meager compared to the potential, and consequently its optimacy has yet to be fully contemplated for the country's prosperity and the wellbeing of its citizens [36,78]. The recent initiative by the government to replace wheat imports throughout the dry season by way of irrigated wheat and other horticultural crops for food self-sufficiency and to supply raw materials for the growing agro-industries suggests that irrigation agriculture is high on the policy agenda (several field visits by the PM to dry season irrigation sites across the country, 2022). The speeches of the Prime Minister to the African Union's 35th ordinary session of the African heads of state on 5 February 2022 (https://au.int/en/speeches/20220205/welcoming-

speech-ethiopian-prime-minister-abiy-ahmed-opening-35th-ordinary-session (accessed on 5 June 2022) demonstrate that wheat irrigation holds much promise and is a practice that should be expanded.

**Supplementary Materials:** The following are available online at https://www.mdpi.com/article/10.3390/w15040803/s1, Figure S1: Rout map where field work (FGD, KII, observation) made in the Central Rift Valley, Ethiopia. Figure S2: Photographs showing the different facets in Centra Rift Valley, namely Lake Abijatas expansion in recent years compared to the previous several years, expansion of horticultural farms mainly for export market, threatening land degradation that include gully, deforestation of acacia woodland and expansion of cultivated land. Figure S3: The Galama mountain area and its surroundings as groundwater recharge zones, and rich biodiversity mountain ecology. Figure S4: Mid-altitude agricultural fields Figure S5: Lower plains as ground water discharge zones and low land irrigated agriculture. Table S1: Lists of participants in the Focus Group Discussions (FGDs) and Key Informant Interviews (KIIs) held from 27 July 2021 to 20 August 2021. This list doesn't include discussions held with Rift Valley Basins Development Office and Oromia Bureau of Water and Energy.

**Author Contributions:** Conceptualization, A.B., G.Z. and J.A.; Methodology, A.B., B.T., G.Z., M.N. and J.A.; Formal analysis, A.B., B.T., A.N.M. and T.A.; Investigation, A.B., B.T., A.N.M. and T.A.; Data curation, A.B., B.T., A.N.M. and T.A.; Writing—original draft, A.B., B.T., A.N.M. and M.N.; Writing—review & editing, A.B., B.T., A.N.M., G.Z., T.A., M.N. and J.A.; Visualization, A.B. and T.A.; Supervision, G.Z, M.N. and J.A; Project administration, G.Z., M.N. and J.A.; Funding acquisition, J.A. All authors have read and agreed to the published version of the manuscript.

**Funding:** This research was supported by the Water Security and Sustainable Development Hub funded by the UK Research and Innovation's Global Challenges Research Fund (GCRF) [grant number: ES/S008179/1] and the Water and Land Resource Centre at Addis Ababa University.

**Institutional Review Board Statement:** Not applicable.

**Informed Consent Statement:** To conduct this research, we got permission from various levels of government offices. This includes permit from (a) Oromia's Regional State Bureau of Water, Energy and Resource Development (Ref No. BMOBI/23/3077, date 24/03/2013 EC), (b) Rift Valley Basin Office and (c) from 30 woredas' departments of agriculture. Participants of the FGD also gave their consent through their signatures on the attendance list. Furthermore, official letters written by Water and Land Resource Centre, Addis Ababa University (Ref. No. WLRC-11/196/2021, date 23 July 2021; WLRC-2/166/202 dated 1 December 2020) helped various stakeholders and institutions to participate in the research either in the interview or data acquisition.

**Data Availability Statement:** All data used in this study are available upon request from the corresponding author.

**Acknowledgments:** The authors would like to thank residents we interviewed at CRV and many other stakeholders who provided us necessary information and data. We also thank several colleagues who have contributed inputs and comments in the development of this chapter. We are also grateful to the anonymous reviewers.

**Conflicts of Interest:** The authors declare that they have no known competing interests that could have appeared to influence the work reported in this paper.

## Appendix A. Government Policy and Strategy Documents and Laws Cited in the Paper

1. GoE (Government of Ethiopia). 2021 Ten Years Development Plan (2021–2030). Amharic version. http://www.pdc.gov.et/#/tenyearplansection (accessed on 25 April 2022).
2. Federal Democratic Republic of Ethiopia, Water Sector Policy (1999)
3. Federal Democratic Republic of Ethiopia, Ethiopian Water Sector Strategy (2001)
4. Federal Democratic Republic of Ethiopia, Rural Development Policy and Strategy (2003)
5. Federal Democratic Republic of Ethiopia, Environmental Policy of Ethiopia (1997)
6. Federal Democratic Republic of Ethiopia, River Basin Councils and Authorities Proclamation No. 534/2007

7. Federal Democratic Republic of Ethiopia, Definition of Power, Duty and Organization of the Basin Development Authority Regulation No. 441/2018
8. Federal Democratic Republic of Ethiopia, Ethiopian Water Resources Management Proclamation No., 197/2000
9. Federal Democratic Government of Ethiopia, Ethiopian Water Resources Management Proclamation No. 197/2000
10. GoE (Government of Ethiopia). (n.d). First Five Year Development Plan (1957–1961). Addis Ababa
11. GoE (Government of Ethiopia). (n.d). Second Five Year Development Plan (1962–1967 E.C). Addis Ababa
12. GoE (Government of Ethiopia) Second Five Year Development Plan (1962–1967 E.C), Third Five Year Development Plan (1968–1973). Addis Ababa

## Appendix B. Summary of the Chronology and Major Objectives of Major Water Institutions Established in Ethiopia to Date

| Year | Name of Institution | Selected Major Objectives |
|---|---|---|
| 1956–1964 | Water Resource Department, Ministry of Public Works, and Communications | Handle a multi-purpose investigation of the Blue Nile Basin. |
| 1962–1977 | Awash Valley Authority (AVA) (Government General Notice No. 299/1962) | To establish plans and programs for the use and development of the resources of the Awash Valley; To coordinate the activities of all Government Ministries and Public Authorities in respect of the use and development of the resources of the Awash Valley; To authorize third parties to construct, acquire, manage, administer, and maintain dams, reservoirs, canals, power houses, power structures, transmission lines and incidental works in the Awash Valley; To administer all water and water rights in the Awash Valley and to control the flow of water of the Awash River. |
| 1971–1975 | National Water Resource Commission (NWRC), Ministry of Public Works and Water Resources (Order No. 75/1971) | To provide full attention to the protection, and efficient utilization and management of all activities relating to water; To ensure the optimum development and use of the nation's inland water resources; To ensure coordination of all activities which may influence the quality, quantity, distribution, or use of water; To ensure appropriate standards and techniques for investigation, use, control, protection, management, and administration of water. |
| 1975–1981 | Ethiopian Water Resource Authority, Ministry of Mines, Energy and Water resources (with three agencies) (proclamation No., 39/1975) | Responsible for: design of water abstraction facilities, water charge collection, irrigation projects O and M |
| 1977–1981 | Valleys Agricultural Development Authority (Proclamation No. 118/1977), under the Ministry of Agriculture and Settlement. Repealed General Notice No. 299/1962) | To study or cause the study of agricultural resources of the river valleys; To prepare and implement plans and programs for the development of and use of agricultural resources in the river valleys; To arrange for the administration, conservation, environmental protection, management and utilization all agricultural resources in the river valleys and to coordinate all the agricultural development activities carried on by different government agencies; To fix and collect fees and charges for the use of water its supplies for agricultural development and for other facilities and services provided by it; to supervise and coordinate the activities of river valley development agencies. |
| 1977–1981 | Awash Valley Development Authority (AVDA) (established under VADA in 1977) | To coordinate the activities of all government and public bodies in respect of agricultural use and developments of resources of the Valley; To conduct studies of agricultural resources of the Valley; To prepare plans and programs for the use and development of agricultural resources in the Valley; To issue directives relating to the use of water for irrigation, land and other facilities, and approve and accept appropriate agricultural practices; In consultation with the Ethiopian Water Resources Authority, to administer all water of the Awash River; To assign water of the valley for irrigation and to fix and collect fees and charges for the use of such water and other facilities; In connection with the directives set by the Authority to design and construct major civil engineering works for the purpose of agricultural development. |

| Year | Name of Institution | Selected Major Objectives |
|---|---|---|
| 1981 | National Water Resource Commission (comprising four agencies including Water supply and Sanitation Authority established) (Proclamation No. 271/1981) | Established as sole authority on the development of the nation's inland and transboundary water resources and responsible for the coordination of meteorological services |
| 1987- | Ethiopian Valleys Development Studies Authority (Proclamation No. 318/1987) | Conduct studies and research of natural resources, in particular water resources, in the valleys of the country; Prepare development masterplans for valleys; Conduct studies and research for the protection of the environment; Conduct studies and research pertaining to transboundary rivers. |
| 1995- | Ministry of Water Resources (Proc. No 4/1995); re-established several times with different names since then (Ministry of Water and Energy- Proc. No. 621/2010; Ministry of Water, Irrigation and Electricity (Proc. No 916/2015; Ministry of Water and Energy- Proc. No. 1263/2021) | Undertakes the management of water resources, water supply and sanitation, large and medium scale irrigation, electricity, and natural and manmade energy resources; Is a regulatory body which involves the planning, development and management of resources, preparation and implementation of guidelines, strategies, polices programs, and sectoral laws and regulations; Conducts study and research activities, provides technical support to regional water and energy bureaus; Engages in the negotiation and the signing of international agreements. |
| 2007 | River Basin Council and Authorities Proclamation No. 534/2007) | The overall objectives of River Basin High Councils and Authorities shall be to promote and monitor the integrated water resources management process in the river basins falling under their jurisdictions with a view to using of the basins' water resources for the socio-economic welfare of the people in an equitable and participatory manner, and without compromising the sustainability of the aquatic ecosystems. |
| 2011 | Rift Valley Lakes Basin High Council and Authority establishment Regulation No. 253/2011 (now amended as Rift Valley Basin Development Office) | The overall objectives of the Authority shall be to promote and monitor the implementation of integrated water resources management process in an equitable and participatory manner in the Rift Valley Lakes Basin. |
| 2018 | Basin Development Authority- Regulation No. 441/2018 | The overall objective of the Authority shall be to implement sustainable and integrated development, administration, and utilization of the water resources at a basin level in equitable and participatory manner. The Authority shall, among other things, facilitate and undertake activities necessary for implementation of integrated water resources management in basins, /ensure that projects, activities, and interventions related to water in the basins are, in line with the integrated water resources management process and/develop plans for protection and sustainable uses of basins; and follow-up implementation once it is approved by the relevant organ |

Note(s): Sources: Consolidated laws of Ethiopia, 1972; Negarit Gazetta of 1971, 1977, 1975, 1981, 1987, 1995, 2007, 2015, 2018, and 2021.

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
