# Peer review of "Voices in Shaping Water Governance: Exploring Discourses in the Central Rift Valley, Ethiopia"

_water, doi:10.3390/w15040803_

Round 1

Reviewer 1 Report

The article is pertinent and contributes definitively to the processes of water security

Author Response

Dear Reviewer;

Thank you so much for your compliments.

kind regards

Reviewer 2 Report

I hate citations in the middle of sentences (such as Salleh 2016) as it is grammatically incorrect. Perhaps weak governance is at the heart of most developing world issues?? Some of these sentences are super long! I think this article would benefit from language editing. Intro is long. First paragraph under 2.1 is not relevant. Much more social, political and economic info on the basin is necessary. 

Author Response

Dear Reviewer,

Thank you for your comments. Kindly we attached our responses to your comments and queries.

very kind regards

Amare Bantider (on behalf of the authors)

Reviewer 3 Report

I enjoyed reading this manuscript. The analysis is scientifically sound and I believe will be of interest to a wide readership, including policy-makers. The main comment I have is that I don't really feel that you draw on political ecology / economy in your analysis, in the sense that you don't really delve into the power relationships between different actors, and explain why some actors gather around specific discourses or why some discourses may gain dominance over others, which actors/voices get invisible in some discourses, which actors benefit/loose in these discourses, etc in your results section. Some of the recommendations you provide also do not really consider or address existing power inequalities (e.g. l. 406-408). You discuss about power relationships in your discussion section, but it remains very abstract and general. I would therefore recommend that you consider some of the above questions if you want to claim that your paper draws on a political ecology approach.

I also have below a few comments to improve the overall paper:

Introduction: I feel that since discourses and narratives are central concepts in your paper, you could provide the definition in the main text, and not as an endnote. Also it would be useful to clarify how you distinguish discourses and narratives (here or in section 2.2). Here or in section 2.1, you could also explain why you chose to focus on the CRV.

Section 2: To me, the end of your section 2.2 starting from l. 182 and and section 2.3 are already part of your results. I would therefore suggest to move it to the results section 3. Could you also clarify how you identified these dominant discourses and why the discourses presented in figure 2 and table 1 don't exactly match (for instance the multiple water use discourse is not in table 1)?

Section 2.3 offers useful contextual factors, it would be useful to add some more details on the timeline of these changes and evolutions (e.g. when did intensive irrigation start, or when has land use started to change, etc)

Discussion: it is really useful to reflect about the relationships/competition between different discourses here but Figure 6 does not show the same discourses than those discussed earlier. Also I think you could reflect on the extent to which some of the discourses in the CRV represent broader national or international discourses / or are specific to the CRV.

Minor comments

l. 68-69: it seems you compare water governance discourses with scholarly works - maybe clarify here "scholarly works analysing water governance discourses and narratives"

l. 86-90: I am not sure it adds a lot to provide a definition of "a system" here as you don't really draw on this concept later in your paper.

l. 140-142: I would nuance this a bit "actors may construct...", as the actors do not always strategically construct and deploy discourses - Hajer (1995) for instance shows that the use of discursive power is not always strategic and intentional.

Figure 2: I am not sure it represents "theoretical frameworks", it rather provides a historical view on dominant discourses 

l. 288-308: I think you don't need to go too much into the details of the Constitution and could reduce this part.

l. 442-443: do you mean to give priority to water use for basic needs / drinking water?

l. 639-640 and 647-650: it is not clear how these two points relate to institutional weaknesses.

Author Response

Dear Reviewer, 

Thank you in deed for your critical comments and suggestions. We greatly benefited for the comments and queries. we, as much possible addressed all comments point by point.

very kind regards

Reviewer 4 Report

Arrange the references in chronological order and in proper format within the text

Author Response

Dear Reviewer;

Thank you so much for reviewing our manuscript. As per the comments of the journal editor we followed the referencing style of the journal.

very kind regards

Amare

Round 2

Reviewer 3 Report

Thank you for considering my comments and for your detailed response.  I feel the manuscript has much improved and is ready for publication. I am looking forward to seeing it online.